# A Sulfur Containing Melanogenesis Substrate, *N*-Pr-4-*S*-CAP as a Potential Source for Selective Chemoimmunotherapy of Malignant Melanoma

**DOI:** 10.3390/ijms24065235

**Published:** 2023-03-09

**Authors:** Yasuaki Tamura, Akira Ito, Kazumasa Wakamatsu, Toshihiko Torigoe, Hiroyuki Honda, Shosuke Ito, Kowichi Jimbow

**Affiliations:** 1Department of Pathology, Sapporo Medical University School of Medicine, South 1 West 16, Chuo-ku, Sapporo 060-8556, Hokkaido, Japan; torigoe@sapmed.ac.jp; 2Department of Chemical Systems Engineering, School of Engineering, Nagoya University, Furo-cho, Chikusa-ku, Nagoya 464-8603, Aichi, Japan; ito.akira@material.nagoya-u.ac.jp; 3Institute for Melanin Chemistry, Fujita Health University, 1-98 Degakugakubo, Kutsukake-cho, Toyoake 470-1192, Aichi, Japan; kwaka@fujita-hu.ac.jp (K.W.); sito@fujita-hu.ac.jp (S.I.); 4Department of Biomolecular Engineering, School of Engineering, Nagoya University, Furo-cho, Chikusa-ku, Nagoya 464-8603, Aichi, Japan; honda@chembio.nagoya-u.ac.jp; 5Institute of Dermatology & Cutaneous Sciences, 1-27 Odori West 17, Chuo-ku, Sapporo 060-0042, Hokkaido, Japan; jimbow@sapmed.ac.jp

**Keywords:** melanogenesis, tyrosinase substrate, novel melanoma immunotherapy, bone marrow-derived dendritic cell, CD8^+^ T cell-dependent immunity

## Abstract

*N*-propionyl-4-*S*-cysteaminylphenol (*N*-Pr-4-*S*-CAP) is a substrate for tyrosinase, which is a melanin biosynthesis enzyme and has been shown to be selectively incorporated into melanoma cells. It was found to cause selective cytotoxicity against melanocytes and melanoma cells after selective incorporation, resulting in the induction of anti-melanoma immunity. However, the underlying mechanisms for the induction of anti-melanoma immunity remain unclear. This study aimed to elucidate the cellular mechanism for the induction of anti-melanoma immunity and clarify whether *N*-Pr-4-*S*-CAP administration could be a new immunotherapeutic approach against melanoma, including local recurrence and distant metastasis. A T cell depletion assay was used for the identification of the effector cells responsible for *N*-Pr-4-*S*-CAP-mediated anti-melanoma immunity. A cross-presentation assay was carried out by using *N*-Pr-4-*S*-CAP-treated B16-OVA melanoma-loaded bone marrow-derived dendritic cells (BMDCs) and OVA-specific T cells. Administration of *N*-Pr-4-*S*-CAP induced CD8^+^ T cell-dependent anti-melanoma immunity and inhibited the growth of challenged B16F1 melanoma cells, indicating that the administration of *N*-Pr-4-*S*-CAP can be a prophylactic therapy against recurrence and metastasis of melanoma. Moreover, intratumoral injection of *N*-Pr-4-*S*-CAP in combination with BMDCs augmented the tumor growth inhibition when compared with administration of *N*-Pr-4-*S*-CAP alone. BMDCs cross-presented a melanoma-specific antigen to CD8^+^ T cells through *N*-Pr-4-*S*-CAP-mediated melanoma cell death. Combination therapy using *N*-Pr-4-*S*-CAP and BMDCs elicited a superior anti-melanoma effect. These results suggest that the administration of *N*-Pr-4-*S*-CAP could be a new strategy for the prevention of local recurrence and distant metastasis of melanoma.

## 1. Introduction

In the past few decades, the prevalence of melanoma has been increasing, leading to increases in morbidity and mortality. Metastatic melanoma is extremely difficult to manage and continues to have a poor prognosis, with an overall 5-year mortality close to 90% [1]. The reason for this poor prognosis is the lack of effective conventional therapies. Various types of therapy used for the management of melanoma have been studied, including immunotherapy, chemotherapy and biologic therapy. Moreover, treatment with immune checkpoint inhibitors and targeted inhibitors of BRAF^V600E^ and MEK kinases has led to significant improvements in the rate of survival and quality of life for a certain proportion of patients in advanced stages [2]. Despite this progress, the benefits of these treatments in the overall population are still marginal, and there is a strong heterogeneity in terms of the percentage of response. Therefore, there is an emerging need for innovative therapies that can be used for the prevention of metastasis, as well as to control advanced melanoma.

The biological property unique to melanoma cells and melanocytes is the biosynthesis of melanin pigments [3]. It is called melanogenesis and occurs within melanosomes. Melanogenesis begins with the conversion of amino acid tyrosine to 3,4-dihydroxyphenylalanine (dopa), and subsequently to dopaquinone in the presence of tyrosinase [4,5]. Therefore, targeting the melanogenesis pathway could be a new strategy for the treatment of malignant melanomas. Tyrosine analog, which is the substrate of the melanin-forming enzyme tyrosinase, may be the best candidate for the development of specific melanoma-targeting drugs and therapies [6,7,8,9]. We previously showed that a sulfur–amine analog of tyrosine, 4-*S*-cysteaminylphenol (4-*S*-CAP), and its *N*-acetyl or propionyl derivatives (NAcCAP or NPrCAP) are good substrates for melanoma-specific targeting and therapy [6,7,10,11,12,13]. They have been shown to cause selective cytotoxicity against melanocytes and melanoma cells after selective uptake into melanoma cells and melanoma transplants, exerting a cytotoxic effect through oxidative stress that may derive from the tyrosinase-catalyzed production of cytotoxic free radicals from the tyrosine analogs [14,15]. The specific cytotoxicities of *N*-Pr-4-*S*-CAP were examined in various types of cultured cells and we found that *N*-Pr-4-*S*-CAP induced apoptotic cell death selectively in melanoma cells, in association with increased caspase 3 activity [13]. Therefore, they can be good candidates for developing anti-melanoma chemotherapy because melanogenesis is inherently toxic and is expressed uniquely in melanocytic cells [7,15,16]. Moreover, we have demonstrated that a putative neoantigen could be generated from *N*-propionyl 4-*S*-cysteaminyl-1-2-benzoquinone (*N*-Pr-4-*S*-CAQ), which is a tyrosinase-activated product of *N*-Pr-4-*S*-CAP, by binding melanosomal proteins through their cysteine residue [17]. These results suggest that *N*-Pr-4-*S*-CAP can trigger the melanoma-specific immune response.

Recently, we synthesized *N*-Pr-4-*S*-CAP/Magnetite (*N*-Pr-4-*S*-CAP/M) nanoparticles, which are new magnetite nanoparticles on which *N*-Pr-4-*S*-CAP was directly conjugated on the surface of the magnetite (Fe_3_O_4_) nanoparticles [5,18,19]. We have shown that *N*-Pr-4-*S*-CAP/M nanoparticles specifically target melanoma cells and are internalized via melanosomes and aggregated within the cell cytoplasm [19]. We have also observed that necrotic cell death occurred in B16 melanoma cells that were subjected to intracellular hyperthermia using *N*-Pr-4-*S*-CAP/M nanoparticles with alternating magnetic field (AMF) exposure, resulting in tumor growth retardation [18,20]. Moreover, we have shown that intracellular hyperthermic treatment using *N*-Pr-4-*S*-CAP/M nanoparticles with AMF exposure induces tumor-specific immune responses, and we have therefore called this anti-melanoma therapy “chemo-thermo-immuno (CTI) therapy” [18,21,22]. As a mechanism, we have shown that CTI therapy-induced anti-melanoma immunity was mediated through cross-presentation of an up-regulated intracellular and extracellular heat shock proteins (HSPs)–antigen peptide complex derived from melanoma cells [21].

In the next step, it is crucial to establish methods for the long-term maintenance of melanoma-specific memory T cells that will be activated when there is a melanoma recurrence, and then find and kill melanoma cells. In our previous study, we found that the *N*-Pr-4-*S*-CAP induced apoptotic cell death in melanocytes, as well as melanoma cells, through the production of reactive oxygen species (ROS) [8,9,11]. Therefore, we examined whether the administration of *N*-Pr-4-*S*-CAP into naïve mice induced anti-melanocyte/melanoma immune responses through melanocyte death. We showed that the intraperitoneal injection of *N*-Pr-4-*S*-CAP induced vitiligo at multiple sites in eight (80%) out of ten C57BL/6 mice. Interestingly, these mice showed anti-melanoma effects against the B16F1 melanoma challenge. The anti-melanoma effect was dependent on CD8^+^ T cells, and isolated CD8^+^ T cells recognized the melanoma-specific antigen, tyrosinase-related protein 2 (TRP2) peptide [13]. These results demonstrated that the administration of *N*-Pr-4-*S*-CAP stimulates a melanoma-specific CD8^+^ T cell response via immunogenic melanoma cell death and can be used as a prevention therapy against the recurrence of melanoma through the maintenance of anti-melanoma immunity after *N*-Pr-4-*S*-CAP-mediated CTI therapy.

Moreover, we showed that the intratumoral injection of *N*-Pr-4-*S*-CAP in combination with bone marrow-derived dendritic cells (BMDCs) exerted a powerful anti-tumor effect and induced very strong anti-tumor immunity. Mice treated with combination therapy through the administration of *N*-Pr-4-*S*-CAP and BMDCs showed complete tumor rejection after a secondary challenge of melanoma cells. Mechanistically, BMDCs took up *N*-Pr-4-*S*-CAP-induced apoptotic melanoma cells, followed by a cross-presentation of melanoma-specific antigen peptides, such as TRP2 to CD8^+^ T cells. Thus, combination therapy using an intratumoral injection of *N*-Pr-4-*S*-CAP and BMDCs against advanced melanoma is a promising strategy not only for the treatment of primary melanoma, but also for prevention of the recurrence of melanoma.

## 2. Results

### 2.1. Administration of N-Pr-4-S-CAP Induced an Anti-Melanoma Immune Response in Naïve Mice

Previously, we demonstrated that an intratumoral injection of *N*-Pr-4-*S*-CAP induced apoptosis in melanocytes and melanoma cells through the increased production of ROS [7,9,10]. Furthermore, we demonstrated that mice injected with *N*-Pr-4-*S*-CAP showed a depigmented lesion on the skin, at a site away from the injection site, through apoptosis of melanocytes. Based on these observations, we considered the possibility that the appearance of a depigmented lesion is a result of cytotoxic T cells (CTLs) killing melanocyte, specifically melanosome-associated antigens through *N*-Pr-4-*S*-CAP-induced melanocyte death. Therefore, we examined whether *N*-Pr-4-*S*-CAP administration elicits systemic anti-melanoma immunity. C57BL/6 mice and nude mice were administered with 100 µg of *N*-Pr-4-*S*-CAP by intraperitoneal injection. Eight of the ten C57BL/6 mice manifested depigmentation. Ninety days later, B16F1 melanoma cells were inoculated into mice that showed depigmentation (NPrCAP group) or naïve mice (PBS group) (Figure 1a). The group of C57BL/6 mice with *N*-Pr-4-*S*-CAP administration showed apparent tumor growth retardation compared to the mice with PBS administration (Figure 1b,c). It is of note that two of the ten mice in the *N*-Pr-4-*S*-CAP group rejected B16F1 cells. In contrast, we did not observe any tumor growth retardation in the nude mice, regardless of *N*-Pr-4-*S*-CAP administration. Based on these results, we assumed that administration of *N*-Pr-4-*S*-CAP induced long-term (at least 90 days) anti-melanoma immunity. Therefore, we next tried to identify the effector cells responsible for anti-melanoma immunity using an in vivo T cell depletion assay. As shown in Figure 2, the depletion of CD8^+^ T cells resulted in the abrogation of *N*-Pr-4-*S*-CAP-induced anti-melanoma immunity. To confirm these results, we examined whether specific anti-melanoma CD8^+^ T cell responses were induced in mice that had rejected B16F1 cells. As a result, B16F1-specific CTLs were induced in mice that had rejected B16F1 cells (Figure 1d). Moreover, the induced CTLs recognized the melanoma-associated antigen peptide TRP2 in the context of H2-K^b^. These results suggested that *N*-Pr-4-*S*-CAP administration elicited anti-melanoma CTL responses through melanocyte death.

### 2.2. Intratumoral Injection of BMDCs in Combination with N-Pr-4-S-CAP Treatment Resulted in Marked Suppression of Melanoma Growth

We have demonstrated the effect of the therapeutic use of *N*-Pr-4-*S*-CAP against established B16 melanoma [8,13]. However, its effect was modest and therefore further improvement was needed. We examined whether DC therapy augmented the anti-tumor effect of *N*-Pr-4-*S*-CAP. On days 10 and 12, *N*-Pr-4-*S*-CAP was intratumorally injected, and then BMDCs (1 × 10^5^) were injected intratumorally on days 12 and 14, after B16F1 inoculation and tumor growth was compared. Although the groups with *N*-Pr-4-*S*-CAP treatment and DC treatment showed growth inhibition (*p* < 0.05), the group with a combination of *N*-Pr-4-*S*-CAP injection and DC administration showed dramatic inhibition of tumor growth (Figure 3). The combination therapy using *N*-Pr-4-*S*-CAP and DCs resulted in the best rate of survival (Figure 4).

### 2.3. Combination Therapy Using N-Pr-4-S-CAP and DCs Elicited Anti-tumor Immunity

To investigate whether combination therapy using *N*-Pr-4-*S*-CAP and DCs elicits anti-melanoma-specific immune responses, we rechallenged B16F1 melanoma in B16F1-rejected mice that were treated with the combination therapy using *N*-Pr-4-*S*-CAP and DCs. Although the mice that were inoculated with B16F1 cells showed progressive tumor growth, the group of mice treated with *N*-Pr-4-*S*-CAP and DCs showed complete rejection of re-challenged B16F1 cells (Figure 5), indicating that the intratumoral injection of *N*-Pr-4-*S*-CAP and DCs induced strong anti-melanoma immunity.

### 2.4. Melanoma-Specific Antigen TYRP2-Specific Cytotoxic T Cells Were Induced in Mice Treated with N-Pr-4-S-CAP and DCs

To determine the mechanism for the generation of anti-melanoma immunity that was caused by an intratumoral injection of *N*-Pr-4-*S*-CAP and DCs, we examined whether a melanoma-specific cytotoxic T cell response was induced in B16F1-rejected mice after combination therapy. Spleen cells of B16F1-rejected mice after combination therapy showed high cytotoxicity against B16F1 melanoma cells compared to the cytotoxicity they showed against EL4 lymphoma cells (Figure 6). The spleen cells also showed high cytotoxicity against EL4 cells pulsed with TRP2 peptide compared to the cytotoxicity they showed against EL4 cells pulsed with human melanoma antigen peptide hgp100 (Figure 6). An ELISPOT assay also showed that TRP-2-peptide specific CD8^+^ T cells were induced (Figure 7). These results suggest that combination therapy administering *N*-Pr-4-*S*-CAP and DCs to melanoma cells can elicit anti-melanoma specific immunity by inducing CTLs against melanoma cells.

### 2.5. BMDCs Phagocytosed N-Pr-4-S-CAP-Treated Melanoma Cells and Cross-Presented Melanoma-Specific Antigen to CD8^+^ T Cells

Finally, we investigated the role of BMDCs in the induction of melanoma-specific CTL responses. In this experiment, we used B16-OVA cells that expressed the surrogate antigen OVA and a B3Z CD8^+^ T cell hybridoma that recognized OVA-derived antigen peptide SL8. *N*-Pr-4-*S*-CAP-treated B16-OVA cells were of a smaller size due to cell death when compared with untreated B16-OVA cells (Figure 8 upper column). BMDCs efficiently phagocytosed *N*-Pr-4-*S*-CAP-treated B16-OVA cells when compared with untreated B16-OVA cells (11.48% vs. 7.31%, respectively) (Figure 8 upper and lower column). BMDCs isolated from coculture with *N*-Pr-4-*S*-CAP-treated B16-OVA cells or untreated B16-OVA cells using CD11c microbeads were then cocultured with B3Z cells. The B3Z cell response against BMDCs loaded with *N*-Pr-4-*S*-CAP-treated B16-OVA cells was evident when compared with the response against BMDCs that were loaded with untreated B16-OVA cells (Figure 9). We therefore concluded that BMDCs take up *N*-Pr-4-*S*-CAP-treated melanoma cells and promoted the cross-presentation of a melanoma-specific antigen(s) to CD8^+^ T cells.

## 3. Discussion

We have extensively demonstrated that CTI therapy using NPrCAP/magnetite and AMF is a promising anti-melanoma therapy [18,19,23]. The features of CTI therapy are as follows: (1) since NPrCAP is a tyrosinase-specific substrate, it is selectively incorporated into melanocytes and melanoma cells through vesicular transport, and adverse effects are therefore expected to be minimal [10,13] and (2) the generation of heat by magnetite nanoparticles with AMF induces not only heat-mediated cell death but also an immune reaction, due to the generation of heat shock proteins (HSPs) [18,20]. HSPs bind endogenous melanoma antigen peptides and they are cross-presented by antigen-presenting cells such as DCs, resulting in the induction of melanoma-specific CTLs [24,25,26,27,28]. Namely, the combination of NPrCAP and magnetite nanoparticles, together with AMF, is considered a novel approach for the development of not only an anti-melanoma pharmacologic agent, but also an immunogenic agent.

The management of metastatic melanoma is an extremely difficult challenge. Currently, only 10% of patients with metastatic melanoma survive for five years because of the lack of effective therapies, even if immune checkpoint inhibitors are used [1,29]. There is, therefore, an emerging need to develop innovative therapies for the control of advanced metastatic melanoma. From another perspective, strategies for the inhibition of metastasis may have greater prospects. To achieve this, it is essential to maintain a high enough level of melanoma-specific anti-tumor immune response to nip metastasis in the bud. We have demonstrated that the administration of *N*-Pr-4-*S*-CAP to mice resulted in melanocyte apoptosis and depigmentation [13]. Furthermore, when *N*-Pr-4-*S*-CAP was added to melanoma cells in culture, apoptosis was induced by increased production of ROS, resulting in cell death [13,30]. Therefore, in the present study, we investigated the possibility of using *N*-Pr-4-*S*-CAP to generate and maintain anti-melanoma immunity, proposing it as a candidate for post-CTI therapy. We found that the administration of *N*-Pr-4-*S*-CAP induced CTLs against TYRP2, a melanocyte-associated antigen, and resulted in tumor growth inhibition after a subsequent melanoma challenge. Importantly, anti-melanoma immunity was shown to last for at least 90 days after *N*-Pr-4-*S*-CAP administration. Our results indicate that the administration of *N*-Pr-4-*S*-CAP after CTI therapy is a promising strategy for maintaining anti-melanoma immunity in order to prevent recurrence and metastasis of melanoma. Moreover, an intratumoral injection of *N*-Pr-4-*S*-CAP and BMDCs exerted a powerful anti-tumor effect and induced very strong anti-tumor immunity. BMDCs took up *N*-Pr-4-*S*-CAP-induced apoptotic melanoma cells, followed by the cross-presentation of melanoma-specific antigen peptides such as TYRP2 to CD8^+^ T cells. Thus, a combination therapy that uses an intratumoral injection of *N*-Pr-4-*S*-CAP and DCs against recurrent melanoma is a promising strategy.

## 4. Materials and Methods

### 4.1. Mice and Cells

Female C57BL/6J mice were obtained from Hokudo (Sapporo, Japan) and used at 4 to 6 weeks of age. Murine melanoma B16F1 and murine thymoma EL4 cell lines were purchased from ATCC and cultured in Dulbecco’s modified Eagle’s medium (DMEM, Thermo Fisher Scientific, Waltham, MA, USA) supplemented with 10% fetal bovine serum (FBS). B16-OVA is a B16F1 melanoma cell line stably transfected with chicken ovalbumin (OVA) cDNA (kindly provided by Dr. Y. Nishimura, Kumamoto University, Kumamoto, Japan). B16-OVA cells were cultured in DMEM supplemented with 10% FBS and 250 μg/mL of hygromycin B (Thermo Fisher Scientific). B3Z is a CD8^+^ T cell hybridoma that expresses LacZ in response to the activation of T cell receptors specific to the SIINFEKL peptide (SL8; OVA-immunodominant peptide) in the context of H-2K^b^ (kindly provided by Dr. N. Shastri, University of California, Berkeley, CA). B3Z cells were cultured in complete RPMI (Thermo Fisher Scientific) supplemented with 10% FBS. BMDCs were generated from the femurs and tibiae of C57BL/6 mice. The bone marrow was flushed out, and the leukocytes were obtained and cultured in complete RPMI1640 with 10% FBS and 20 ng/mL GM-CSF (Endogen, Inc., Woburn, MA, USA) for 5 days. On day 3, fresh medium with GM-CSF was added to the plates for the 5-day cultures.

### 4.2. Peptides

SL8 (SIINFEKL), TRP2 (SVYDFFVWL), and hgp100_25-33_ (KVRPNQDWL) were synthesized on a solid phase support using F-moc for transient NH_2_ terminal protection and characterized using mass spectrometry. They were purified by HPLC to >99% homogeneity and stored at 2 mM in distilled H_2_O_2_ at −80 °C.

### 4.3. Preparation of N-Pr-4-S-CAP (NPrCAP)

*N*-Pr-4-*S*-CAP (MW = 225 D) was kindly provided by Alberta University’s Department of Dermatology (Canada). The compound was synthesized as described by Tandon et al. [8]. For the study, *N*-Pr-4-*S*-CAP was dissolved in propylene glycol (Wako, Osaka, Japan) at a concentration of 244 mM and sterilized by filtration.

### 4.4. Animal Models for Tumor Formation

Female C57BL/6J mice (4 weeks old, approximately 10.0 g) were obtained from Hokudo (Sapporo, Japan). The mice were injected with 100 μg of *N*-Pr-4-*S*-CAP or propylene glycol (100 μL) into an intraperitoneal lesion. Ninety days after injection, 3.0 × 10^5^ B16F1 melanoma cells in 0.1 mL of phosphate-buffered saline (PBS) were injected subcutaneously into the right flank of each of the C57BL/6J mice. Tumor diameters were measured every other day. In another experiment, 3.0 × 10^5^ B16F1 melanoma cells in 0.1 mL of PBS were injected subcutaneously into the right flanks of C57BL/6J mice on day 0. On day 10, 20 mice were randomly divided into four treatment groups. On days 10 and 12, tumor-bearing mice were injected with 0.1 mL of *N*-Pr-4-*S*-CAP (24.4 mmol, 5.5 mg) in propylene glycol or 0.1 mL of propylene glycol alone directly into the tumor with a 26-gauge micro syringe or propylene glycol alone. Immature BMDCs (1 × 10^5^) were injected intratumorally on days 11 and 13. Tumor diameters were measured every other day.

Animal experiments were carried out according to the principles described in the “Fundamental Guidelines for Proper Conduct of Animal Experiments and Related Activities in Academic Research Institutions under the jurisdiction of the Ministry of Education, Culture, Sports, Science and Technology” of Japan.

### 4.5. In Vivo T Cell Depletion Assay

Mice were injected with 100 μg of *N*-Pr-4-*S*-CAP or propylene glycol (100 μL) into intraperitoneal lesion. Ninety days after injection, mice were depleted of CD4^+^ or CD8^+^ T cells by an intraperitoneal injection of 200 g of GK1.5 (purchased from ATCC) ascites supernatant or 2.43 (purchased from ATCC) ascites supernatant, respectively, on days 3, 4, 11 and 18. The amount of depletion was confirmed by flow cytometry, and specific depletion was more than 95% in each group. Mice in the antibody control group were injected with 200 g of rat IgG (Dainippon Sumitomo Pharmaceutical, Osaka, Japan). On day 0, 3.0 × 10^5^ B16F1 melanoma cells in 0.1 mL of PBS were injected subcutaneously into the right flank of each of the C57BL/6J mice. Tumor diameters were measured every other day.

### 4.6. In Vitro Cytotoxicity Assay

After the mice had been treated with *N*-Pr-4-*S*-CAP and DC combination therapy, spleen cells were harvested from the mice that had been completely cured by the combination therapy by day 42. Five 10^6^ spleen cells were then re-stimulated with irradiated B16F1 cells in 2 mL of RPMI1640 supplemented with 50 μM β-mercaptoethanol (Invitrogen, Carlsbad, CA, USA) and 10% FBS for five days. The cytotoxic activity of spleen effector cells against target B16F1 cells, EL4 cells, EL4 cells coated with TRP-2 peptide (2 μg/mL) or YAC-1 cells was determined by standard ^51^Cr release assays.

### 4.7. ELISPOT Assay

Spleen cells were removed on day 42 from mice that had been completely cured by the combination therapy. The cells were stimulated with TYRP-2 peptide (2 μg/mL) for five days. CD8^+^ T cells were isolated with MACS (Miltenyi Biotec, Bergisch Gladbach, Germany) using an anti-mouse CD8a mAb coupled with magnetic microbeads according to the manufacturer’s instructions. As target cells, EL4 cells were cultured overnight at 26 °C in RPMI 1640 supplemented with 10% FBS and 100 μg/mL TYRP-2 peptide or hgp100 peptide or without any peptide. Ninety-six-well ELISPOT plates (BD Bioscience, Franklin Lakes, NJ, USA) were coated with 5.0 μg/mL rat anti-mouse IFN-γ mAb and subsequently blocked with RPMI 1640 supplemented with 10% FBS for 2 h at room temperature. Then 5 × 10^3^ cytotoxic T lymphocytes (CTLs) and 1 × 10^5^ of each of the target cells were added to the wells and cultured for 12 h at 37 °C in RPMI 1640 with 10% FBS. The plates were then washed extensively and incubated with 2.0 μg/mL of biotinylated anti-mouse IFN-γ mAb, followed by pulsing with 0.5 μg/mL streptavidin-HRP. Positive spots were counted using a Vision ELISPOT reader (Carl Zeiss, Oberkochen, Germany).

### 4.8. BMDC Phagocytosis Assay

B16-OVA cells were preloaded with the dye PKH-red (1 mg/mL, Invitrogen/Molecular Probes) for 1 h at 37 °C. BMDCs were preloaded with 5 μg/mL of the dye PKH-green (1 kDa; Invitrogen/Molecular Probes) for 1 h at 37 °C. B16-OVA cells were layered over BMDCs at a ratio of 1:3. The cells were incubated at 37 °C for 6 h. B16-OVA cells phagocytosed by BMDCs were observed by using a fluorescence microscope (Radiance 2000MP, BIO-RAD, Hercules, CA, USA) and the percentage of the phagocytosed cells was calculated as the fraction of BMDCs that accepted B16-OVA cells (yellow/orange cells) in the total population of BMDCs (yellow/orange and green cells) using FACS vantage (Becton, Dickinson and Company (BD), Franklin Lakes, NJ, USA).

### 4.9. In Vitro Cross-Presentation Assay

BMDCs (5 × 10^5^ cells) from C57BL/6 mice were loaded with B16-OVA cells (1 × 10^4^ cells) or *N*-Pr-4-*S*-CAP-treated B16-OVA cells (1 × 10^4^ cells) for 2 h at 37 °C in 100 μL of Opti-MEM. Then, BMDCs were purified using CD11c MACS beads (Miltenyi Biotec). Purified BMDCs were plated at a density of 1 × 10^3^/200 μL in 10% RPMI and cocultured overnight with 1 × 10^5^ B3Z cells. BMDCs loaded with SL8 (1 μM) served as a positive control. Stimulated B3Z cells were stained with CPRG (Roche) and red color was measured as absorbance at 595 nm.

### 4.10. Statistical Analyses

The data were analyzed by one-way or two-way analysis of variance (ANOVA) and then differences in experimental results for tumor growth were assessed by Sheffe’s test to compare all of the experimental groups or by Dunnett’s test for the experimental groups vs. the control group. For multiple comparisons, the data were assessed by the log-rank test with Bonferroni correction. The differences in survival rates were analyzed by the Kaplan–Meier method. The level of significance was *p* < 0.05 (two-tailed). All statistical analyses were performed using StatView J-5.0 (SAS Institute Inc., Cary, NC, USA).

## 5. Conclusions

*N*-Pr-4-*S*-CAP is a substrate for a melanin biosynthesis enzyme, tyrosinase, which is highly and uniquely expressed in malignant melanoma. This study examined the mechanism responsible for the selective induction of *N*-Pr-4-*S*-CAP-mediated anti-melanoma immunity by (a) cell depletion assay to identify effectors cells and (b) cross-presentation assay to clarify the involvement of CD8^+^ T cells in melanoma cell death. *N*-Pr-4-*S*-CAP administration can be a new immunotherapeutic approach against melanoma, including local recurrence and distant metastases.

## Figures and Tables

**Figure 1 ijms-24-05235-f001:**
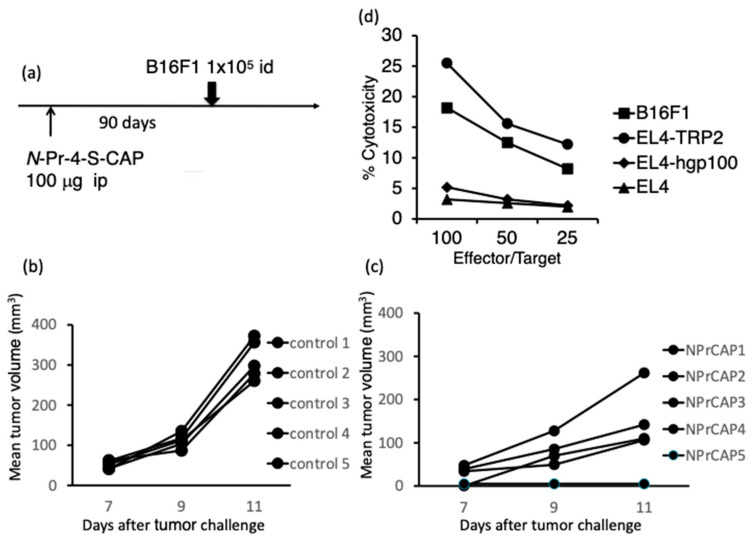
Administration of *N*-Pr-4-*S*-CAP induces anti-melanoma immunity and melanoma-specific CTLs in naïve mice. (**a**) Naïve mice were administered with *N*-Pr-4-*S*-CAP (100 μg in 100 μL propylene glycol/mice) or propylene glycol (100 μL/mice) by intraperitoneal injection. About six weeks after the *N*-Pr-4-*S*-CAP injection, mice that showed vitiligo or white hair were chosen. (**b**,**c**) Ninety days after the *N*-Pr-4-*S*-CAP injection, mice were transplanted intradermally with B16F1 cells (1 × 10^5^ cells) and tumor diameters were measured. (**d**) Splenocytes from mice that had rejected B16F1 melanoma in *N*-Pr-4-*S*-CAP-treated group showed B16F1-specific and melanoma antigen (TRP2)-specific CTL activity.

**Figure 2 ijms-24-05235-f002:**
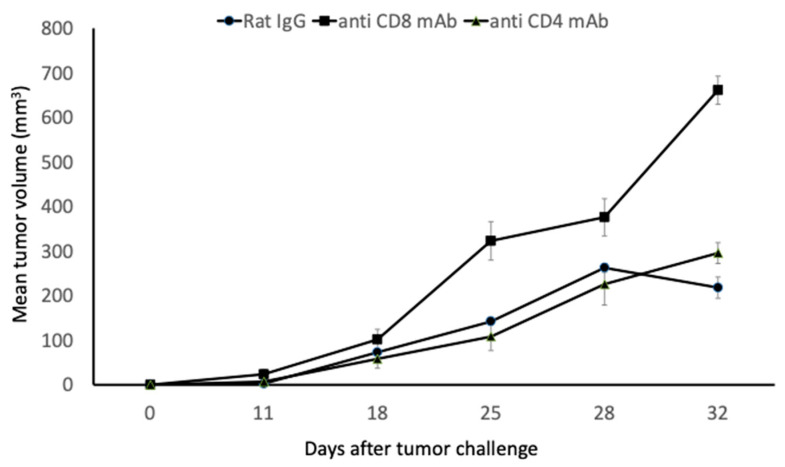
CD8^+^ T cells are effector cells for *N*-Pr-4-*S*-CAP-induced anti-melanoma immunity. Mice were injected with 100 μg of *N*-Pr-4-*S*-CAP or propylene glycol (100 μL into intraperitoneal lesion). Ninety days after injection, mice were depleted of CD4^+^ or CD8^+^ T cells by an intraperitoneal injection of 200 μg of GK1.5 ascites supernatant or 2.43 ascites supernatant, respectively, on days −3, 4, 11 and 18. Mice in the antibody control group were injected with 200 μg of rat IgG. On day 0, 3.0 × 10^5^ B16F1 melanoma cells in 0.1 mL of PBS were injected subcutaneously into the right flank of each of the C57BL/6J mice. Tumor diameters were measured every other day.

**Figure 3 ijms-24-05235-f003:**
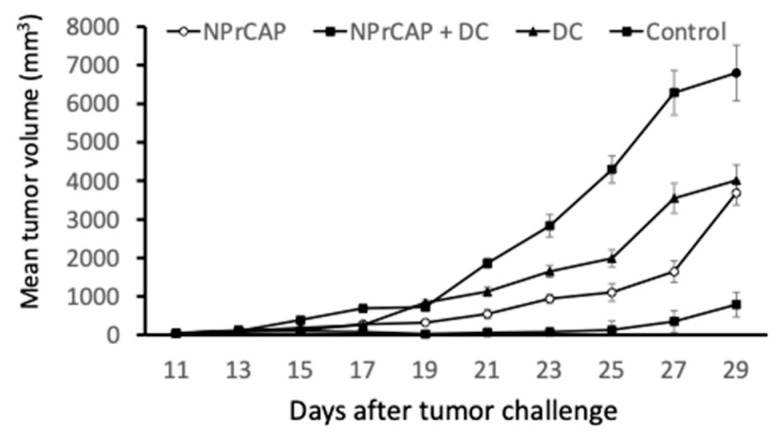
*N*-Pr-4-*S*-CAP inhibits B16F1 melanoma growth and combination therapy with an intratumoral injection of BMDCs augments the anti-tumor activity of *N*-Pr-4-*S*-CAP anti-tumor. Mice were transplanted with B16F1 cells and treated with *N*-Pr-4-*S*-CAP, BMDCs and *N*-Pr-4-*S*-CAP in combination with BMDCs. Mice bearing B16F1 melanomas were treated with 24.4 μmol *N*-Pr-4-*S*-CAP, BMDCs (1 × 10^5^) or *N*-Pr-4-*S*-CAP and BMDCs by intratumoral injections on days 11 and 13. Tumor diameters were measured every other day.

**Figure 4 ijms-24-05235-f004:**
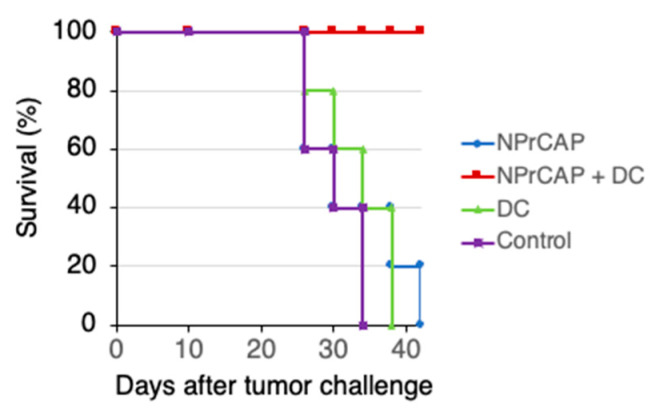
Combination therapy using *N*-Pr-4-*S*-CAP and BMDCs resulted in prolongation of the survival periods of B16F1-bearing mice. Kaplan–Meier survival curves after tumor transplantation showed significant prolongation of survival in the combination therapy (*N*-Pr-4-*S*-CAP and BMDCs) group compared to the control group (*p* < 0.05).

**Figure 5 ijms-24-05235-f005:**
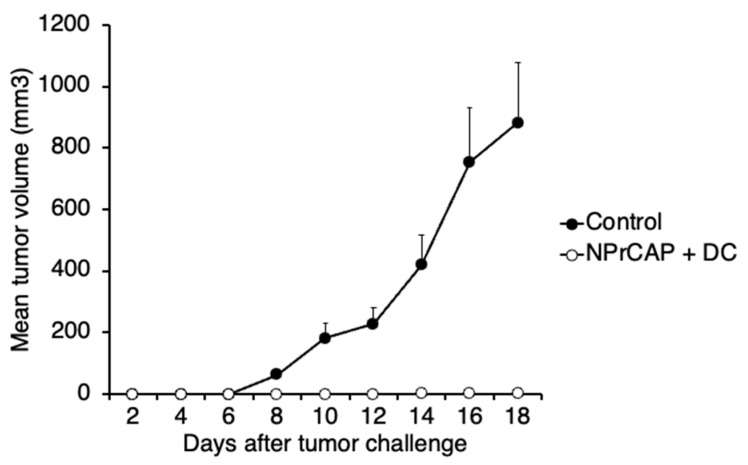
Growth inhibition of a secondary transplanted tumor in mice in the *N*-Pr-4-*S*-CAP and BMDCs treatment group. Mice that had been cured by *N*-Pr-4-*S*-CAP and BMDCs combination therapy were rechallenged with B16F1 melanoma, and tumor growth was compared to that in naïve mice transplanted with B16F1 melanoma.

**Figure 6 ijms-24-05235-f006:**
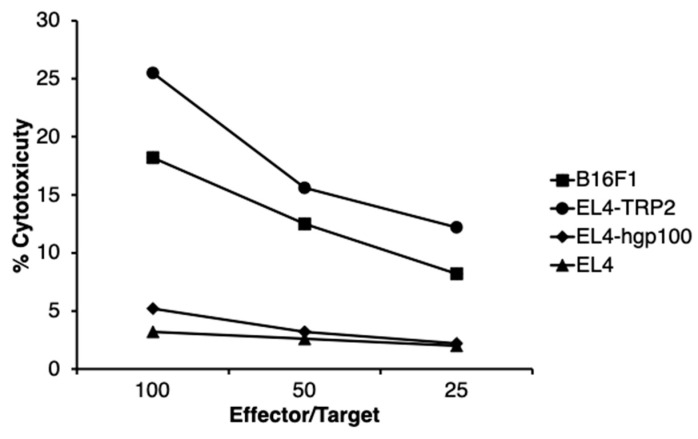
Induction of CTLs against B16F1 cells and TRP2 peptide by combination therapy that used intratumoral administration of *N*-Pr-4-*S*-CAP and BMDCs as determined by ^51^Cr release assay. Mice were treated with *N*-Pr-4-*S*-CAP and BMDCs combination therapy, and then spleen cells were harvested from mice that were completely cured and re-stimulated with irradiated B16F1 cells for 5 days. Cytotoxic activity of spleen effector cells against B16F1cells, EL4 cells, EL4 cells coated with TRP-2 peptide or YAC-1 cells was determined by a standard ^51^Cr release assay.

**Figure 7 ijms-24-05235-f007:**
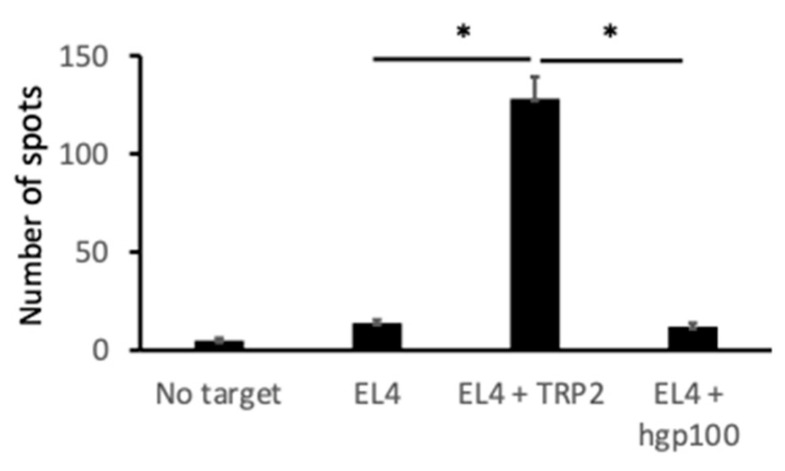
Induction of CTLs against B16F1 cells and TYRP2 peptide by combination therapy that used intratumoral administration of *N*-Pr-4-*S*-CAP and BMDCs as determined by ELISPOT assay. Mice were treated with combination therapy (*N*-Pr-4-*S*-CAP and BMDCs). Spleen cells from cured mice were stimulated with TRP-2 peptide (2 μg/mL) for five days. CD8^+^ T cells were isolated with MACS using an anti-mouse CD8a mAb coupled with magnetic microbeads. As target cells, EL4 cells were loaded with TRP2 peptide or hgp100 peptide or without any peptide. ELISPOT assay using IFN-γ was carried out. *, *p* < 0.05.

**Figure 8 ijms-24-05235-f008:**
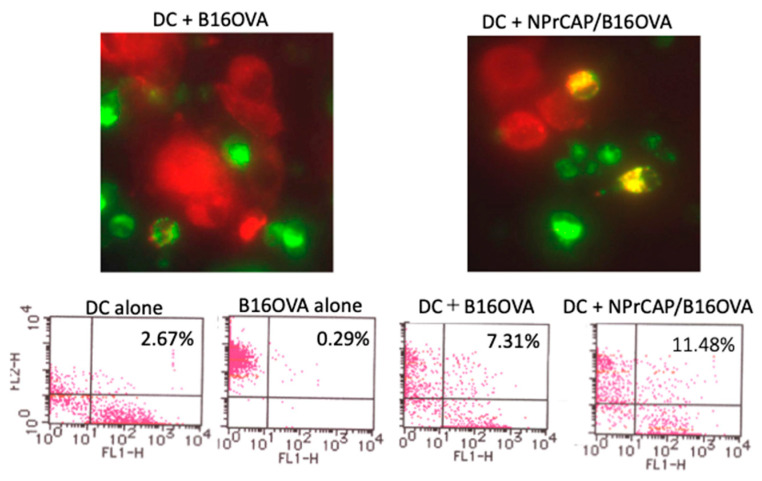
BMDCs take up *N*-Pr-4-*S*-CAP-treated B16-OVA melanoma cells. PKH-red-labeled B16-OVA cells treated with/without *N*-Pr-4-*S*-CAP were cocultured with PKH-green-labeled BMDCs for 6 h. Phagocytosed B16-OVA cells were then analyzed using fluorescent microscopy (original magnification × 100), upper column) and a flow cytometry (lower column).

**Figure 9 ijms-24-05235-f009:**
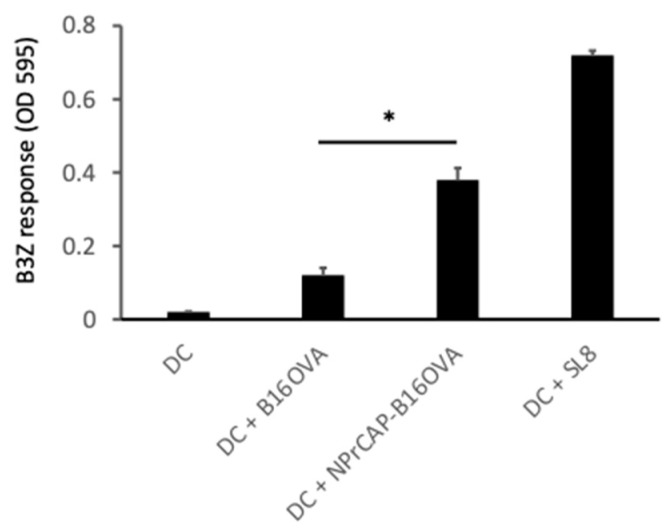
BMDCs cross-present a melanoma-associated antigen to antigen-specific B3Z CD8^+^ T cells. BMDCs were cocultured with B16-OVA or *N*-Pr-4-*S*-CAP-treated B16-OVA cells for 6 h. Then, BMDCs were isolated using CD11c microbeads and cocultured with OVA-specific T cell hybridoma B3Z cells. BMDCs loaded with SL8 peptide served as a positive control. *, *p* < 0.05.

## Data Availability

Data sharing is not applicable to this article.

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
