# Peer review of "A Sulfur Containing Melanogenesis Substrate, N-Pr-4-S-CAP as a Potential Source for Selective Chemoimmunotherapy of Malignant Melanoma"

_ijms, 2023, doi:10.3390/ijms24065235_

Round 1
Reviewer 1 Report
In the manuscript the authors presented their finding, that intratumoral injection of N-Pr-4-S-CAP in combination with bone marrow derived dendritic cells had a powerful antitumor effect and induced very strong antimelanoma immunity mediated by cytotoxic T cells. They showed that DC take up N-Pr-4-S-CAP-treated B16-OVA melanoma cells and stimulate antigen specific CD8 T cells. Taken together, the authors conclusively demonstrated, that N-Pr-4-S-CAP administration can be a new immunotherapeutic approach against melanoma. This is a relevant paper that can be accepted in the current form.
Author Response
Thakn you very much for reviewing our manuscript and we are grateful for your comments and decision.
Reviewer 2 Report
Congratulations to the authors for their work.
A suggestion for the future would be to expand their molecular study to determine if N-Pr-4-S-CAP inhibits all melanoma cells, regardless of the presence of genetic mutations.
Author Response
Thank you very much for reviewing our manuscript and giving us an important suggestion. We agree with your comment. We are planning to compare the effect of N-Pr-4-S-CAP using humaan melanoma cell lines with/without BRAF V600E mutation.